# Anaemia in Children and Adolescents: A Bibliometric Analysis of BRICS Countries (1990–2020)

**DOI:** 10.3390/ijerph18115756

**Published:** 2021-05-27

**Authors:** Olushina Olawale Awe, Dennis Makafui Dogbey, Ronel Sewpaul, Derrick Sekgala, Natisha Dukhi

**Affiliations:** 1Department of Mathematical Sciences, Anchor University Lagos, Lagos 100278, Nigeria; oawe@aul.edu.ng; 2Institute of Infectious Diseases and Molecular Medicine, Faculty of Health Sciences, University of Cape Town, Cape Town 7701, South Africa; DGBDEN001@myuct.ac.za; 3Human and Social Capabilities Division, Human Sciences Research Council, Merchant House, 116-118 Buitengracht Street, Cape Town 8001, South Africa; rsewpaul@hsrc.ac.za (R.S.); dsekgala@hsrc.ac.za (D.S.)

**Keywords:** anaemia, adolescents, BRICS, authors, publications, research collaborations

## Abstract

Background/objectives: Brazil, Russia, India China and South Africa have prioritized cooperation regarding health, including malnutrition. Anaemia in children, adolescents and women of reproductive age has been on the increase in these countries, placing a huge strain on healthcare systems. This study aimed to map the scientific perspective and research publications on anaemia in children and adolescents in the BRICS countries. Methods: Bibliometric analyses were employed to map scientific publications related to anaemia in children and adolescents in BRICS countries using VOSviewer software. Research documents from 1990 to 2020 were imported from PubMed. Descriptive statistics was used to analyse trends in research publications, authorship and keywords over the 30-year period. Findings: BRICS countries accounted for 15% of all publications on the subject within the last three decades. While India had the highest number of publications, China had the author with the highest number of research publications and co-authorship links. Of all article types, India had the highest number of letters, while China and South Africa published the highest number of RCT/clinical trial and review articles, respectively. Conclusion: The review of all scientific studies on anaemia in BRICS nations for the past 30 years revealed gaps in research collaborations on anaemia between authors in BRICS nations. However, collaborative research projects may contribute to building a shared base of evidence, innovations, data and methodologies for a more comprehensive understanding of the risks and vulnerabilities of child and adolescent anaemia. This will aid in the development and evaluation of interventions and policies to alleviate anaemia and nutrient deficiencies.

## 1. Background/Introduction

Anaemia is a major public health concern, affecting young children, adolescents and women of child-bearing age. According to the World Health Organization (WHO), an estimated 42% of children under the age of five years and 40% of pregnant women globally are anaemic [1]. Anaemia is characterized by a lower-than-normal blood haemoglobin level, resulting in substandard motor and cognitive development in children and loss of productivity in work during adulthood. A common cause of anaemia is iron deficiency, as well as other micronutrient deficiencies such as folic acid, vitamins B12 and A, genetic haemoglobin disorders and infectious diseases [2]. Young children are at particularly high risk of anaemia due to increased iron and other micronutrient demand required for growth, as well as having a higher infection susceptibility [3]. Anaemia in children has led to poor socio-emotional, cognitive and psychomotor development, which can result in irreversible and long-lasting impacts on growth and development, even if iron deficiency is rectified [4]. Adolescents, defined by the WHO as those aged 10–19 years [5] are in an important transitional period of life between child- and adulthood, characterized by increased growth and development, that offers a second and final chance for growth catch up as part of the life cycle [6,7]. Thus, the micro and macronutrient need among adolescents are higher due to this rapid growth and development period [8]. Rapid growth, inadequate dietary iron intake and menstrual blood loss place females at particular risk of developing iron deficiencies [9]. Given the crucial growth periods as children transition into adulthood, there is a need for science to advance nutrition and health research in these important population groups.

Such nutrition and health related research advancements are developing among Brazil, Russia, India, China and South Africa, commonly known as the BRICS nations, who may not yet have formal political alliance but do have prioritized cooperation regarding health. A recent study found that despite this health cooperation, India and South Africa had the poorest healthcare system performance, with the authors identifying strong correlations between mortality and health expenditure, funding, and human resources deficiencies, as well as magnifying the activities to reduce unhealthy lifestyles in BRICS countries as prioritization points to improve health system outcomes [10]. With regard to anaemia in BRICS countries, in children under the age of five years, the prevalence from 2009 to 2016 appears to be increasing or remaining constantly high. Such is the case in India and South Africa, where the prevalence is consistent over this period, but India ranks as the most concerning, with the highest overall anaemia prevalence. In women of reproductive age, anaemia prevalence is somewhat similar to child prevalence, with India and South Africa having decreased prevalence from 2009 to 2016 but still contributing the most to anaemia prevalence in comparison to the other BRICS nations [11,12]. 

Iron-deficiency anaemia is directly linked to malnutrition and poverty. Children are likely to become stunted or wasted due to poverty and inability to consume well-balanced diets; women are unable to access iron and folic tablets during pregnancy; and poor sanitation, malaria occurrences and worm infestations also result in high occurrences of anaemia [13]. Thus, there is a need for these countries to increase funding in terms of healthcare if anaemia in these countries is to be effectively managed and controlled. Research plays an integral part in assisting scientists, policymakers and other stakeholders in understanding how we advance science through collaborations and scientific cooperation, and it is important to have up to date information on the trends of health research and research productivity. This can be complemented by the bibliometric analysis approach undertaken in this article.

Bibliometric analysis is a methodology applied to quantitatively measure scientific publications to determine the research productivity of journals [14,15]. In bibliometric analysis, there are several descriptive statistics that can be taken into account, such as citation data, network analysis that includes journals, authors, countries, keywords, and academic and research institutions, based on citations and utilization of frequency analysis. Furthermore, it can provide insights into research clusters, current interests, and trends in emerging topics [16]. Research publications are regarded as one of the most tangible science production units for any given country [17]. Bibliometric analysis affords researchers and related stakeholders an opportunity to gain a thorough understanding of a field of study and promotes interdisciplinary collaboration [18]. 

Child and adolescent nutrition remain an important public health concern globally. As previously mentioned, young children and adolescents are at greater risk of anaemia due to the increased iron and other micronutrient demand, required for growth and protection against susceptibility to infections [1,3]. A scientific collaboration, focusing on nutrition (anaemia) and machine learning, amongst researchers from BRICS nations in 2020 aimed to conduct a bibliometric analysis of anaemia in children and adolescents in BRICS countries between 1990 and 2020, in order to investigate the research productivity output of researchers/scientists and stakeholders in BRICS countries on child and adolescent Anaemia in published articles included in the PubMed electronic database.

## 2. Methodology

We conducted bibliometric analysis on research conducted by BRICS (Brazil, Russia, India, China and South Africa) member countries on Anaemia in children and adolescents between January 1990 and December 2020. We retrieved published documents from PubMed by using new and varied search terms ((“anaemia” [All Fields] OR “anemia” [MeSH Terms] OR “anemia” [All Fields] OR “anaemias” [All Fields] OR “anemias” [All Fields]) AND (“child” [MeSH Terms] OR “child” [All Fields] OR “children” [All Fields] OR “child s” [All Fields] OR “childrens” [All Fields] OR “childrens” [All Fields] OR “childs” [All Fields]) AND (“adolescences” [All Fields] OR “adolescency” [All Fields] OR “adolescent” [MeSH Terms] OR “adolescent” [All Fields] OR “adolescence” [All Fields] OR “adolescents” [All Fields] OR “adolescents” [All Fields]) AND [Brazil OR Russia OR India OR China OR South Africa]). These terms were built based on previous publications about anaemia in children and adolescents. PubMed was our database of choice as it allows for basic and advanced search options. In our search, we included all publications including books, journal articles, letters and clinical trials to capture all information on the subject. Where articles have authors from two or more member countries, we designated such articles to the country of the first author. Our search yielded 15,345 articles for publications across the world from which 10,000 were imported, 13,055 articles (10,000 imported for analysis) from non-BRICS countries and 2298 publications from BRICS countries. To ensure quality assurance, articles were double checked to eliminate repetitive articles.

The PubMed outputs were imported, from which we generated an excel spreadsheet for each BRICS country. From this, bar charts and graphs were produced to describe trends in publications, authors with highest publication and keywords that were most used in each country and collectively. The same was also undertaken for BRICS countries combined against non-BRICS countries to assess trends in publications. Additional analysis was performed using VOSviewer version 1.6.15 to visualise publications on the subject by BRICS countries. VOSviewer was selected because it is easily accessible, free, and user-friendly.

## 3. Results

In the results below, we analysed the number of publications, author’s keywords, article types and research organizations for each BRICS country and collectively. 

Figure 1 shows the trends in publications of documents by BRICS and non-BRICS countries and globally from 1990 to 2020. It could be seen that there is a general increase in the number of publications over the years. Figure 1 also compares the number of publications by BRICS verses non-BRICS countries. It could be deduced that BRICS countries contributed approximately 15% of all global publications on the subject over the last three decades (January 1990–December 2020), the highest number of publications were in 2019 (not taking into account 2020) and the lowest number of publications occurred in 1991. India had the highest with 926 (36.5%) of the total publications, followed by China with 914 (36.1%), Brazil with 512 (20.2%), South Africa with 208 (8.2%) and Russia with 47 (1.9%), in that order.

From Figure 2, it can be seen that there is a general increase in the number of publications over the last three decades. Specifically, there was an increase (spike) in the number of publications from India between 2004 and 2008 and again between 2010 and 2013. In addition, there was an increase (spike) in the number of publications from China between 2010 and 2018. 

Interestingly, the number of publications from South Africa has generally remained the same in the first two decades until 2011. Meanwhile, the number of publications from Brazil has significantly increased over the years, with an increase of over 200% between 2008 and 2014. Comparatively, Brazil and South Africa recorded a spike in the number of publications compared to India and China in the year 2017. 

### 3.1. Authorship Analysis

The top 10 authors in each country, with at least two published documents overall within the three decades, were analysed. For each of the authors, the total co-authorship link with other authors was calculated. Authors with the greatest total strength were selected and were ranked before considering the number of publications. The results are shown in the figures below. 

It should be noted that the total number of publications per author is the sum of the number of publications depicted as blue and co-authorship links in brown. 

Figure 3a–e shows the top 10 authors with the highest number of publications with corresponding authorships from Brazil, Russia, India, China and South Africa, respectively.

From the figures, the authors with the highest publications are Bonfim Carmen with 18 publications among Brazilian authors, Zamran Ari with four publications among Russian authors, Chandra, Jagdish with 22 publications among Indian authors, Fu Rong with 34 publications among Chinese authors, and Wonkam Ambroise with nine publications among South African authors. 

Interestingly, it could be seen that the number of publications is not directly proportional to the number of co-authorship links. For example, although Wonkam Ambriose from South Africa had nine publications, there were 11 co-authorship links, compared to Rabie Helena with only three publications but as many as 21 co-authorship links. The same dynamics are seen among Indian authors where Chandra Jagdish had 22 publications and 25 co-authorships links compared to George Biju who had 11 publications and 41 co-authorships links. In addition, consider Cipolotti Rosanna from Brazil with eight publications without a co-authorship link; however, Silva Gisele Sampaio, with six publications, had a total of 11 co-authorships links. From this, it could be deduced that there is a poor collaboration among some authors and that authors with higher publications tend to collaborate with wide range of other authors. 

Table 1 shows the top 10 authors with the highest number of publications from the BRICS countries which is dominated by Chinese and Indian authors. It depicts the authors with 15 publications or more on the subject with corresponding co-authorship links within the last 3 decades. Fu Rong had 34 publications with the highest co-authorship links strength of 56 among the BRICS authors. Other authors from BRICS countries with the highest publications are Bonfim Carmen, Zimram Ari, Chandra Jagdish, and Fu Rong and Phirim Sam, representing Brazil, Russia, India, China and South Africa, respectively.

In addition, we analysed research collaborations between authors from these five countries, as displayed in Figure 4 below. We set a threshold of at least one publication to determine co-authorship links or collaboration networks within and between other authors among the BRICS countries. Of the 13,320 authors, 256 publications meet this threshold. 

From the layout algorithm displayed in the figure below, it appears there is weaker or no collaboration between authors across the BRICS countries compared to within a country. 

### 3.2. Authors Keywords Analysis

The minimum number of occurrences of an author’s keyword is five; for Russia it is one. For each of the author’s keywords, the total strength of the co-occurrence link with other keywords was calculated. The top five authors’ keywords with corresponding total link strength were selected and tabulated as represented in the graphs below (Figure 5).

### 3.3. Analysis of Article Types

Article types were determined from PubMed manually for all countries. For each BRICS country, each article type was sourced and counted from the larger search results, tabulated, and plotted as shown below.

Figure 6 shows types of articles published by authors in BRICS countries from 1990 to 2020. Approximately 97% (2229 out of 2298) of all publications were journal articles while book chapters were the least published over the 3 decades. While South Africa published the highest number of review articles (25 out of 87), the highest number of journal articles and letters were published by China and India, respectively. China published the highest number of RCT/Clinical Trials (80 out of 197) and Russia the least (9 out of 197).

### 3.4. Analysis of Research Organisations

For each organisation from BRICS, the highest number of published documents and the corresponding total strength of the co-authorship links with other organizations were calculated. The organisations with the highest number of publications were ranked first and those with the greatest total link strength were selected, as shown in the graphs below. 

Documents with authors > 25 were excluded. The minimum number of documents of an organisation in a country was set to two. For BRICS combined, minimum number of an organisation set to one to allow all organisation. For each organisation, the total strength of the co-authorship links with other organizations was calculated. The organisations with the greatest total link strength were selected. 

Figure 7 a–f shows the organisations with the highest number of publications from Brazil, Russia, India, China, South Africa, and BRICS combined, respectively. From the figures above, the top five organisations with the highest publications in each country and combined were obtained. Among the BRICS countries, Indian organisations represented by the All Indian Institute of Medical Sciences published the highest number of documents (18), followed by the Chinese organisations represented by Peking University First Hospital and the Institute of Haematology and Blood Diseases Hospital, each with seven published documents. The South African Medical Research Council in Cape Town published the fifth highest number of documents within the last 3 decades among BRICS country organisations, and was also the organisation with the highest number of collaborations with other organisations in South Africa.

In terms of collaborations among these top five organisations, which are depicted here in terms of total link strength, no collaboration was found between and within the BRICS countries when set to a minimum of five documents per organisation, as seen in the graphs above. This means that no two BRICS organisations have collaborated on five or more publications within the past 3 decades. However, collaborations existed between and within BRICS countries if the minimum number of documents per organisation was set to four or less. The majority of these collaborations were between Chinese and Indian organisations.

A network analysis of the collaboration between a total of 5812 BRICS organisations which published within the period of 3 decades was conducted. From this number, 1000 were selected and analysed. To allow for fair representation of all organizations, the minimum number of documents per organization was set to one for the VOSviewer software analysis. 

From the figure above, it could be seen that there are generally poor collaborations between BRICS organizations and weak interconnections in terms of the subject matter and publications during the period. 

Figure 8 shows BRICS organizations with at least three published documents within the last 3 decades. Out of a total of 4157 organizations, 49 meet this threshold and the majority of these organizations are Chinese and Indian organizations.

## 4. Discussion

This study used bibliometric data to report on published research on anaemia in children and adolescents in BRICS countries between 1990 and 2020. Overall, 2298 studies from BRICS countries were found, contributing to 15% of the total studies on the subject worldwide. The results were presented according to (i) the number of publications over time, (ii) most published authors, (iii) keywords, (iv) article types, and (v) research organizations, for each BRICS country and collectively.

The overview bibliometric analysis conducted on BRICS countries indicated that there has been a rapid increase in the number of publications, with the highest in India and the least in Russia, since 2000. After 2006, there was a significant rise in the number of publications from China, Brazil and South Africa. Notably, China’s rate of publications grew after 2010, as described by Xie and Freeman (2019) [19], with 37 percent of global citations to scientific articles published in 2013 attributed to China. Additionally, with shares of articles and citations being more than twice its share of global population or GDP, China has achieved a comparative advantage in terms of knowledge. This has tremendous implications on the direction of research and development (R&D) in the world.

Furthermore, our analysis found that between 1990 and 2012, there was no change in the publication rate in Russia. However, this finding cannot be taken at face value when taking into account the fact that the Russian population has always been reporting a significant lower anaemia prevalence as compared to populations from other countries with a similar socio-demographic index [20]. An increase (spike) in the number of publications from India between 2004 and 2008 in this analysis was also reported in the worldwide prevalence of anaemia: WHO global database on anaemia [21]. Several publications from South Africa have generally remained the same within the first two decades until 2011. However, the number of publications from Brazil has significantly increased over the years, with an increase of over 200% between 2008 and 2014. This could be attributed to the fact that in low middle-income countries (LMIC) like South Africa, resources for research funding were not ample, technology was missing, and knowledge dissemination was not appropriate and widespread [22]. 

The study also touched upon the evolution of anemia research in BRICS countries based on keywords and most published authors from the publications. Certainly, published research on these topics contributes significantly to the knowledge of the scientific community and leads to better understanding of the research areas that most require further contributions; however, it could be seen that the number of publications is not directly proportional to the number of co-authorship links, which was also observed in other bibliometric analysis studies [23]; as illustrated in the current analysis, Wonkam Ambriose from South Africa had nine publications, with 11 co-authorship links compared to Rabie Helena with only three publications and 21 co-authorship links. The same dynamics are seen among Indian authors where Chandra Jagdish shown 22 publications and 25 co-authorships links compared to George Biju who had 11 publications and 41 co-authorships links. Nonetheless, the current results suggest that collaborations among BRICS countries occur far less frequently than collaborations between a BRICS affiliate and a high-income, low-burden country including the U.S. and/or U.K, despite the increase in research output over the last 3 decades. On the other hand, north–south collaborations allow for the sharing of valuable expertise and resources to increase collaboration; this trend should be followed to increase collaboration among BRICS countries. 

The most used keywords varied by member country. The co-occurrence link between keywords was generally low. Keywords relating to sickle cell disease tended to cluster separately from the other clusters of keywords. Furthermore, sickle cell disease and sickle cell anaemia were among the top keywords in studies from Brazil.

The majority of studies were journal articles followed by randomized control trials or clinical trials. While China and India far exceeded the other member countries in their absolute numbers of journal articles, the relative percentage of journal articles published within each country were similar. South Africa’s studies comprised a higher proportion of reviews than the other countries. Russia, despite having the lowest absolute number of RCTs and clinical trial studies, had a higher proportion of these studies compared to the other countries.

The top five research organizations with respect to the number of publications comprised two organizations from India, followed by two from China and one from South Africa. The Chinese and Indian organizations with the most publications generally had fewer collaborations with other organizations in their respective countries, while within-country collaborations were more prevalent among the top ranking South African, Brazilian and Russian organizations. 

The findings from this bibliometric analysis provide evidence that China and India have dominated in terms of numbers of publications over the past three decades. Furthermore, the inter-country collaborations have mostly involved organizations from these two countries. However, there was minimal collaboration involving organizations from the other BRICS countries. There are, therefore, opportunities for increased collaboration between BRICS countries on anaemia research in children and adolescents. 

### Study Limitations

This study included documents published only in English, indexed and retrieved from PubMed. It is likely that the conclusion from this analysis may not reflect the actual publication trends as affected by BRICS member countries, organisations and authors whose publications were in languages other than English and have not been indexed or appeared in PubMed.VOSviewer warning: PubMed data on organisations may not have been harmonized. Organization names may not have a consistent format. Other observation was that publications by organisations and authors outside BRICS countries where the published documents included study site and data from BRICS member countries were difficult to eliminate from the PubMed search result/output.

## 5. Conclusions

In this study, a comprehensive bibliometric analysis has been undertaken on research conducted by BRICS (Brazil, Russia, India, China and South Africa) member countries on Anaemia in children and adolescents between January 1990 and October 2020. Published research and review articles were retrieved from PubMed using new and varied search words. A significant amount of interesting and revealing findings were gleaned from this study. It could be seen that there is a general increase in the number of publications over the years. While South Africa published the highest number of review articles (25 out of 87), the highest number of journal articles and letters were published by China and India, respectively. China published the highest number of RCT/Clinical Trials (80 out of 197) and Russia the least (9 out of 197). It is also of note that no two BRICS organisations have collaborated on five or more publications within the past three decades.

### Recommendation and Future Research

The BRICS countries have shared health priorities, including the alleviation of all forms of malnutrition such as undernourishment and micronutrient deficiencies. The ninth BRICS Health Ministers meeting held in Brazil in 2019 reaffirmed the importance of BRICS countries investing and collaborating in research and development activities in various spheres of healthcare and wellbeing [24]. Research provides the evidence underpinning this health policy. All BRICS countries can benefit from increased research collaborations in the area of anemia research in children and adolescents. Collaborative research projects contribute to building a shared base of evidence, innovations, data and methodologies to obtain a more comprehensive understanding of the risks and vulnerabilities of child and adolescent anaemia and to develop and evaluate interventions and policies to alleviate anaemia and nutrient deficiencies. Given that India, Brazil and South Africa have high anaemia prevalence, this is compounded as well by high poverty, unequal societies, and under resourced public health systems. Given the association between the social determinants of health and malnutrition and nutrient deficiencies such as anaemia, these countries are likely to have shared risk factors for anaemia in children and adolescents, which can also be uncovered and addressed through collaborative research networks. Finally, more research funds should be dedicated to conducting research on this burning topic.

## Figures and Tables

**Figure 1 ijerph-18-05756-f001:**
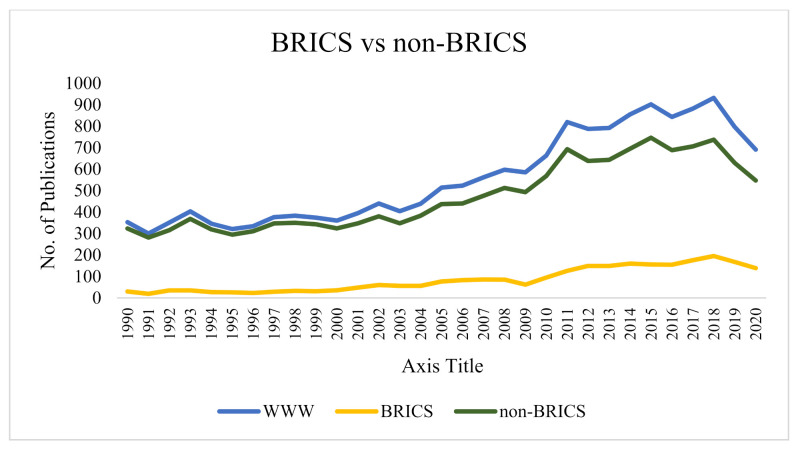
Comparison of publications by BRICS versus non-BRICS countries (1990–2020).

**Figure 2 ijerph-18-05756-f002:**
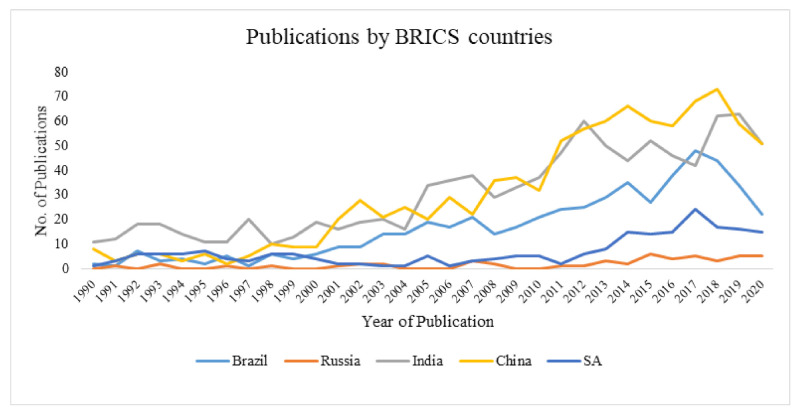
Publication trends by BRICS countries (1990–2020).

**Figure 3 ijerph-18-05756-f003:**
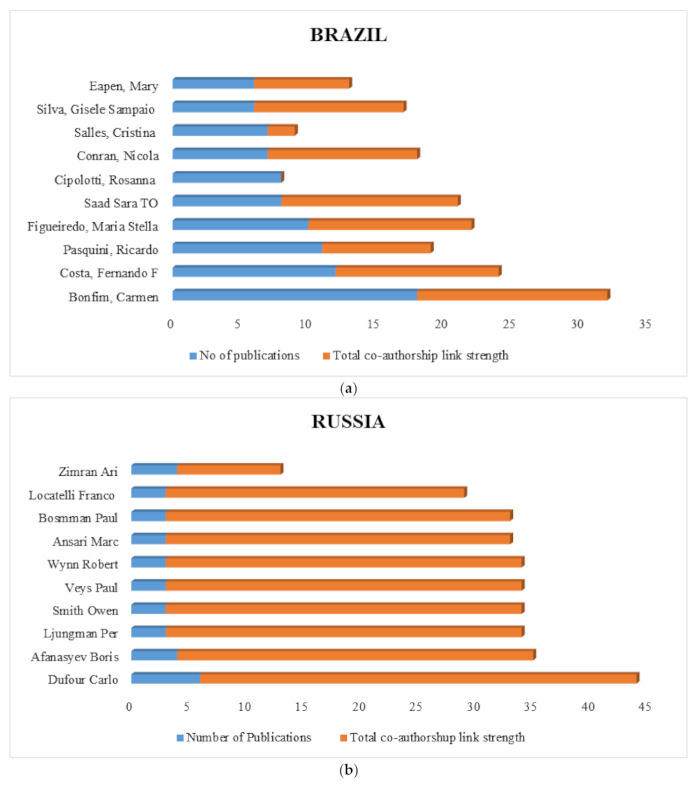
(**a**): Top ten authors from Brazil with at least 5 publications within the 3 decades covered in this study. (**b**) Top ten authors from Russia with at least 3 publications within the 3 decades covered in this study. (**c**) Top ten authors from India with at least 12 publications within the 3 decades covered in this study. (**d**) Top ten authors from China with at least 15 publications within the 3 decades covered in this study. (**e**) Top ten authors in South Africa with at least 4 publications within the 3 decades covered in this study.

**Figure 4 ijerph-18-05756-f004:**
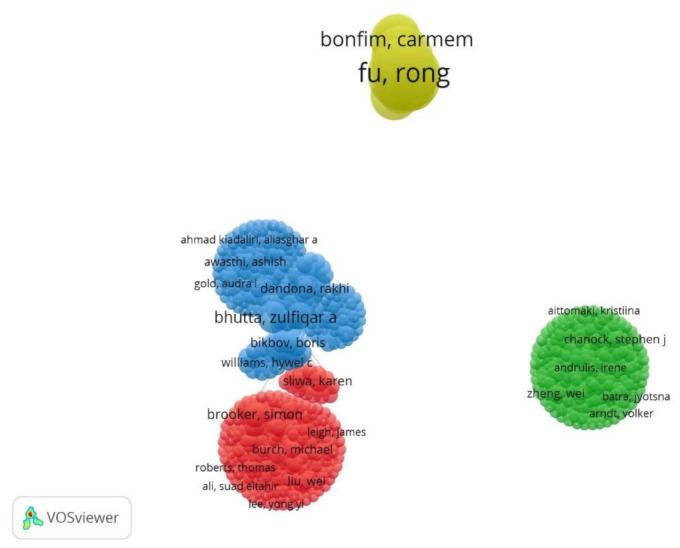
Collaboration networks between BRICS authors.

**Figure 5 ijerph-18-05756-f005:**
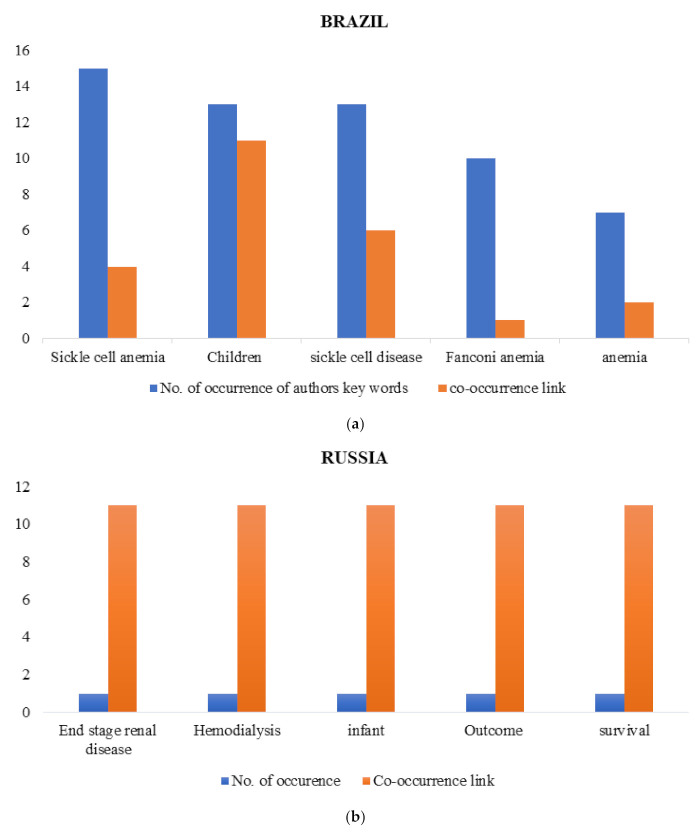
(**a**) Top author keyword analysis: Brazil. (**b**) Top author keyword analysis: Russia. (**c**) Top author keyword analysis: India. (**d**) Top author keyword analysis: China. (**e**) Top author keyword analysis: South Africa. The graphs (**a**–**e**) show the topmost keywords in relation to the search terms as used by authors from Brazil, Russia, India, China and South Africa respectively. The minimum number of the occurrences of a keyword was 5. For each country, the occurrence of a keyword and its corresponding co-occurrence link with other keywords were analysed, as shown in the figures above., It could be seen that wide range of keywords have been used by BRICS authors and with minimal co-occurrence links between these keywords.

**Figure 6 ijerph-18-05756-f006:**
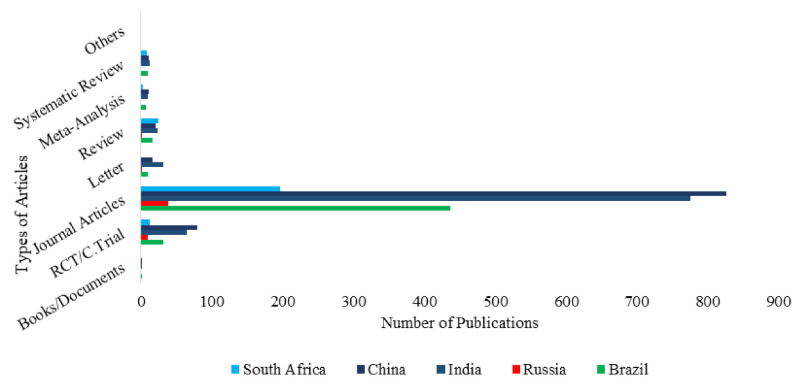
Article types: comparison among BRICS countries.

**Figure 7 ijerph-18-05756-f007:**
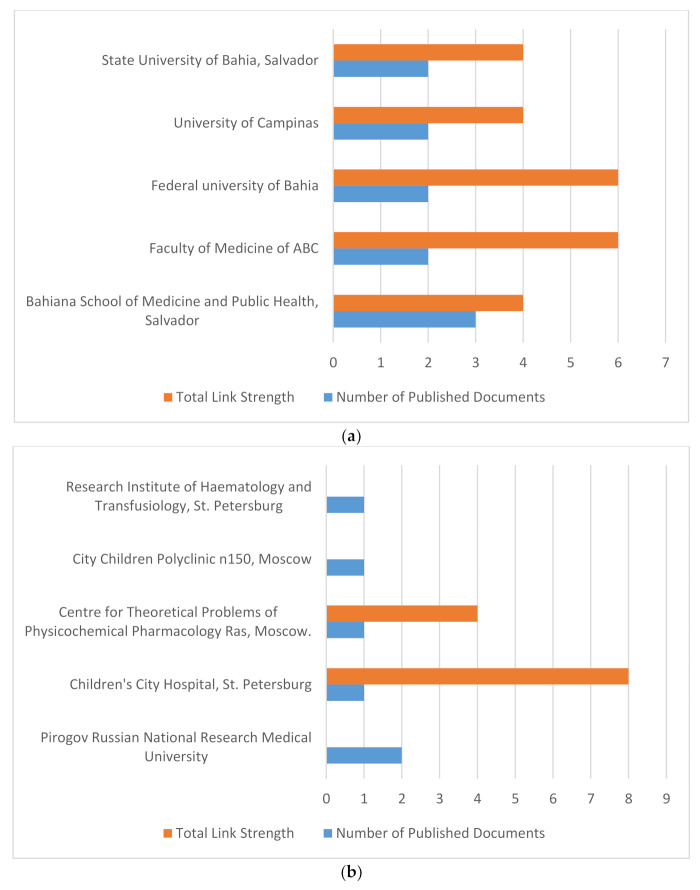
(**a**) Organisations with highest number of anaemia publications: Brazil. (**b**) Organisations with highest number of anaemia publications: Russia. (**c**) Organisations with highest number of anaemia publications: India. (**d**) Organisations with highest number of anaemia publications: China. (**e**) Organisations with highest number of anaemia publications: South Africa. (**f**) Organisations with highest number of anaemia publications: BRICS.

**Figure 8 ijerph-18-05756-f008:**
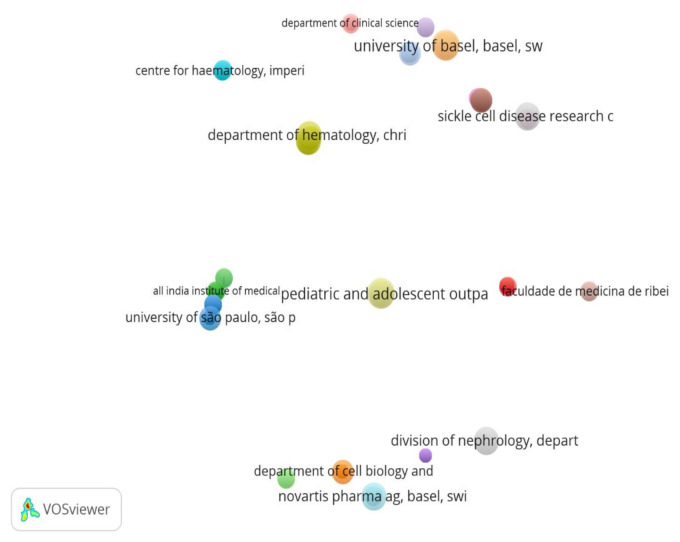
BRICS organizations with at least three published documents within the last three decades.

**Table 1 ijerph-18-05756-t001:** Ranking of the top ten authors in BRICS countries.

Rank	Author	Country of Origin	Number of Publications	Total Co-Authorship Link
1	Fu, Rong	China	34	56
2	Liu, Hong	China	24	51
3	Shao, Zong-Hong	China	20	40
4	Shi, Jun	China	27	33
5	Zheng, Yizhou	China	20	12
6	Choudry, VP	India	18	4
7	Saxena, Renu	India	20	3
8	Chandra, Jagdish	India	23	1
9	Ghosh, Kanjaksha	India	18	1
10	Bonfim, Carmen	Brazil	18	0

## Data Availability

Not applicable.

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
