# Peer review of "Anaemia in Children and Adolescents: A Bibliometric Analysis of BRICS Countries (1990–2020)"

_ijerph, 2021, doi:10.3390/ijerph18115756_

Round 1
Reviewer 1 Report
Please check the attached file.

Reviewer 2 Report
The paper from Awe et al is a bibliometric analysis of the research production in BRICS countries in the last 30 years, focusing on anaemia in the pediatric population. The paper is clear and well written however some points could be improved before publication:
- The period of analysis should be updated to the end of the 2020 (31 dec 2020), now it is Oct 2020, this will allow to have a complete view also of the 2020 and to take a picture of the impact of Covid19 pandemic on the research production in 2020.
- Pediatric/paediatric should be included in the search terms
- It’s not clear what is the meaning of the author keyword analysis and how the terms were chosen and combined (for example why sickle cell anemia and sickle cell disease are not clustered together?...),from my point of view it would be much more informative to cluster the papers based on the most common studied diseases related to anemia in pediatric/adolescent populations (ie sickle cell anemia, iron deficiencies, thalassemia, Fanconi anemia…), I don’t think the listed keywords add any further information.
- What are the error bars in fig 6a referring to?
-
The paper from Awe et al is a bibliometric analysis of the research production in BRICS countries in the last 30 years, focusing on anaemia in the pediatric population. The paper is clear and well written however some points could be improved before publication:
- The period of analysis should be updated to the end of the 2020 (31 dec 2020), now it is Oct 2020, this will allow to have a complete view also of the 2020 and to take a picture of the impact of Covid19 pandemic on the research production in 2020.
- Pediatric/paediatric should be included in the search terms
- It’s not clear what is the meaning of the author keyword analysis and how the terms were chosen and combined (for example why sickle cell anemia and sickle cell disease are not clustered together?...),from my point of view it would be much more informative to cluster the papers based on the most common studied diseases related to anemia in pediatric/adolescent populations (ie sickle cell anemia, iron deficiencies, thalassemia, Fanconi anemia…), I don’t think the listed keywords add any further information.
- What are the error bars in fig 6a referring to?
- It’s not clear how the organizations were chosen, they should be the most productive organizations of a specific country but in many cases there are just one or 2 publications in 30 years of research, these small data don’t seem to fit with the country productions graphed above. If the reason is that many other organizations published just one or two articles the authors should clearly explain why they listed some and left out others.
Round 2
Reviewer 1 Report
the authors revised the manuscript according to the comments. It was acceptable.
Reviewer 2 Report
The authors adequately addressed all the points.